# Bacteria Broadly-Resistant to Last Resort Antibiotics Detected in Commercial Chicken Farms

**DOI:** 10.3390/microorganisms9010141

**Published:** 2021-01-09

**Authors:** Jared M. Jochum, Graham A. J. Redweik, Logan C. Ott, Melha Mellata

**Affiliations:** 1Department of Food Science and Human Nutrition, College of Agriculture and Life Science, Iowa State University, Ames, IA 50011, USA; jmjochum@iastate.edu (J.M.J.); gredweik@iastate.edu (G.A.J.R.); ott@iastate.edu (L.C.O.); 2Interdepartmental Microbiology Graduate Program, Iowa State University, Ames, IA 50011, USA

**Keywords:** ESBL-producing *Enterobacteriaceae*, colistin resistant *Escherichia coli*, carbapenem-resistant *Acinetobacter*, hens, maturity stages, plasmid

## Abstract

Resistance to last resort antibiotics in bacteria is an emerging threat to human and animal health. It is important to identify the source of these antimicrobial resistant (AMR) bacteria that are resistant to clinically important antibiotics and evaluate their potential transfer among bacteria. The objectives of this study were to (i) detect bacteria resistant to colistin, carbapenems, and β-lactams in commercial poultry farms, (ii) characterize phylogenetic and virulence markers of *E. coli* isolates to potentiate virulence risk, and (iii) assess potential transfer of AMR from these isolates via conjugation. Ceca contents from laying hens from conventional cage (CC) and cage-free (CF) farms at three maturity stages were randomly sampled and screened for extended-spectrum β-lactamase (ESBL)-producing *Enterobacteriaceae*, carbapenem-resistant *Acinetobacter* (CRA), and colistin resistant *Escherichia coli* (CRE) using CHROMagar™ selective media. We found a wide-spread abundance of CRE in both CC and CF hens across all three maturity stages. Extraintestinal pathogenic *Escherichia coli* phylogenetic groups B2 and D, as well as plasmidic virulence markers *iss* and *iutA*, were widely associated with AMR *E. coli* isolates. ESBL-producing *Enterobacteriaceae* were uniquely detected in the early lay period of both CC and CF, while multidrug resistant (MDR) *Acinetobacter* were found in peak and late lay periods of both CC and CF. CRA was detected in CF hens only. *bla_CMY_* was detected in ESBL-producing *E. coli* in CC and CF and MDR *Acinetobacter* spp. in CC. Finally, the *bla_CMY_* was shown to be transferrable via an IncK/B plasmid in CC. The presence of MDR to the last-resort antibiotics that are transferable between bacteria in food-producing animals is alarming and warrants studies to develop strategies for their mitigation in the environment.

## 1. Introduction

The rise of antimicrobial resistant (AMR) bacteria is a serious threat to human and animal health, as increasing resistance to commonly used antibiotic therapies have created a burden on treatment options [1]. AMR bacteria can arise in nature and are commonly found in food producing animals like poultry [2]. For instance, the gastrointestinal tract (GIT) of chickens and the facilities that house these poultry serve as reservoirs for AMR resistant bacteria [3,4]. Of these AMR bacteria, extended-spectrum β-lactamase (ESBL) producing *Enterobacteriaceae*, carbapenem-resistant *Acinetobacter* (CRA), and colistin resistant *E. coli* (CRE) are emerging AMR bacteria found in the poultry environment [5,6,7]. The Centers for Disease Control and Prevention (CDC) lists ESBL-producing *Enterobacteriaceae* and CRA as serious and urgent threats, respectively [8]. Although not explicitly listed as an antibiotic resistant threat by the CDC, colistin resistance is clinically relevant given its use as a last resort antibiotic for treating multidrug resistant (MDR) infections [9]. As poultry is one of the most consumed meat sources globally [10], it is crucial to identify the presence and prevalence of these AMR populations in the poultry environment and understand the spread of these resistances to other bacteria.

Bacteria primarily acquire AMR genes by horizontal gene transfer (HGT), a leading contributor to bacterial coevolution [11]. Conjugative plasmids are responsible for HGT of virulence and AMR genes, which has led to the rapid rise of AMR in bacterial pathogens [12,13]. Recently, the transfer of mobile colistin resistance (*mcr*) and ESBL-producing genes have been linked to a variety of plasmid types and bacterial hosts in the poultry environment [14,15]. As AMR populations persist in the poultry environment, there is an increased risk that pathogens might acquire AMR genes.

Although *E. coli* and *Acinetobacter* are commensal gut bacteria in poultry and are detected in the feed, feces, and environment of poultry facilities, both *E. coli* and *Acinetobacter* have the potential to cause extraintestinal diseases in both humans and poultry [16,17,18,19,20,21,22]. Importantly, extraintestinal pathogenic *E. coli* (ExPEC) infections are often highly fatal in humans and poultry and are increasing worldwide, imposing a major burden on public health [23]. As pooling of these potential pathogens and AMR genes are taking place in poultry, it is important to detect both AMR genes and associated virulence markers that can identify potential pathogens. There are limited studies that investigate the role of production environments (i.e., conventional cage [CC] versus cage-free [CF]) and maturity stages (i.e., early, peak, and late lay) on AMR emergence. In commercial farms, layer hens can be categorized by the period in which egg production begins (early), is at its highest (peak), and later diminishes due to age (late). Studies have shown that the maturity stage of layer hens can impact the colonization and shedding of particular bacteria and the diversity of bacteria inhabiting the GIT [24,25]. Recently, our study has shown that different layer maturity stages exhibit differing levels of *Enterobacteriaceae* in CC and CF conditions [26]. We thus hypothesized that both environment and maturity may play a role in AMR diversity and potential virulence detection.

In this study, we examined ceca contents from hens in commercial CC and CF environments as potential reservoirs for CRA, CRE, and ESBL-producing *Enterobacteriaceae*. Potential spread of these resistances was examined as well as the virulence potential of *E. coli* isolates. We were able to identify MDR *Acinetobacter*, ESBL-producing *E. coli*, and widespread presence of CRE in both CC and CF environments. Phylogenetic and virulence screening identified possibly MDR ExPEC isolates in both environments. Finally, AMR was demonstrated to be transferable in the CC environment via plasmid mediated HGT.

## 2. Materials and Methods

### 2.1. Source Material

Ceca contents from hens in two commercial farms (CC and CF) located in Iowa were previously sampled, flash frozen, and stored at −80 °C between May and September of 2017 [24]. Samples were collected from three laying stages: (i) early lay (17–23 weeks), (ii) peak lay (25–39 weeks), and (iii) late lay (64–88 weeks) from the CC and CF commercial farms in Iowa. A total of 20 hens were randomly sampled for each maturity group in each facility (N = 20 × 3 maturities × 2 facilities = 120 total). The samples were analyzed as described in (Figure 1) and detailed below.

### 2.2. Identification of Last Resort Antibiotic-Resistant Isolates

Initially, 0.2 g ceca content samples were resuspended in 200 μL of sterile PBS, and 20 μL of suspensions were then added to CHROMagar COL-APSE™ (Paris, France) media for the selection of colistin-resistant bacteria. For enrichment of AMR bacteria, original suspensions were mixed with 500 μL Luria-Bertani broth (0.1% glucose) and incubated overnight at 37 °C. Thereafter, 20 μL of enriched suspensions were pipetted onto CHROMagar Orientation™ agar (Paris) to detect ESBL-producing *Enterobacteriaceae* and CHROMagar SuperCARBA™ (Paris) to detect carbapenem-resistant bacteria. Plates were incubated at 37 °C for up to 48 h to allow for the growth of slow growing bacteria like *Acinetobacter* spp. [27]. Initially, isolates were speculatively speciated by color change on the CHROMagar™ media according to manufacturer’s instructions. Colistin resistant bacteria were enumerated, and cultures from individual colonies were stored in glycerol solution at −80 °C for future experiments. Colistin resistance was confirmed by growth on Mueller-Hinton agar (MHA) (4 mg/L colistin). Colonies from CHROMagar OrientationTM were later tested on CHROMagar COL-APSE™ and SuperCARBA™ to evaluate multiple resistance. Colonies from CHROMagar SuperCARBA™ were later tested on CHROMagar COL-APSE™ and Orientation™ respectively to evaluate multiple resistance. DNA was extracted from all colistin resistant and ESBL-producing bacteria as described [28] and were screened via polymerase chain reaction (PCR) for the *E. coli* housekeeping gene, *uidA* (Appendix A). White/cream colonies isolated from CHROMagar Orientation™ and SuperCARBA™ media were speculatively designated as *Acinetobacter* spp., DNA was extracted, and isolates were confirmed by PCR amplification of the *Acinetobacter* spp. specific 16S rRNA gene (Appendix A) [29]. All positively identified CRE, ESBL-producing *E. coli* and *Acinetobacter*, and CRA were thereafter screened for resistance to rifampicin (100 μg/mL), tetracycline (15 μg/mL), nalidixic acid (30 μg/mL), ampicillin (50 μg/mL), kanamycin (50 μg/mL), gentamicin (20 μg/mL), and chloramphenicol (20 μg/mL). β-lactamase production was confirmed through resistance to third generation cephalosporins (cefotaxime and ceftazidime) via Clinical and Laboratory Standards Institute (CLSI) guidelines [30]. Furthermore, minimum inhibitory concentration (MIC) of colistin, cefotaxime, and ceftazidime were obtained via agar dilution using CLSI breakpoints [28]. For all antibiotic resistance assays, *E. coli* K-12 strain MG1655 was used as a negative control. PCRs were performed to identify the presence of various *mcr* genes, multiple β-lactamase encoded genes, and carbapenemase encoded genes (Appendix A). ESBL and *mcr* multiplexes were utilized as described previously [31,32]. NCBI Primer-blast was used to generate *bla_TEM_* primer pairs (https://www.ncbi.nlm.nih.gov/tools/primer-blast/). Positive control strains used in this portion of the study were supplied from the CDC & FDA Antibiotic Resistance Bank (Appendix A).

### 2.3. Plasmid Profiling and Typing

CRE, ESBL-producing *E. coli* and *Acinetobacter*, and CRA isolates were streaked onto LB agar plates and incubated overnight. Individual colonies from overnight plates were suspended in 3 mL LB broth (0.1% glucose) and shaken overnight at 37 °C. Plasmid extraction was performed using phenol: chloroform-based method as described previously [33]. The resulting plasmid extracts were loaded into 0.5% TAE agarose gels for agarose gel electrophoresis. Gels were run overnight at 40 V and 4 °C, stained with ethidium bromide, and imaged using a c300 imager (Azure Biosystems). Approximate band sizes were calculated using GelAnalyzer software (GelAnalyzer 19.1). All CRE and ESBL-producing *E. coli* were subjected to plasmid replicon typing for the detection of 18 common *E. coli* incompatibility groups (IncB/O, FIC, A/C, P, T, K/B, W, FIIA, FIA, FIB, Y, I1, Frep, X, HI1, N, HI2, and L/M) via multiplex PCR [34].

### 2.4. PCR Screenings for ExPEC

PCRs were performed to identify the common *E. coli* phylogenetic groups (A, B1, B2, and D) and the presence of virulence factors associated with ExPEC plasmids (*cvaC*, *iroN*, *iss*, and *iutA*) (Appendix A).

### 2.5. AMR Transfer Assays

To assess horizontal gene transfer of plasmids carrying AMR genes by our isolates, we selected ESBL-producing *E. coli* isolates from CC (IA-EC-0010, IA-EC-0018) and CF (IA-EC-0075, and IA-EC-0076) as donors, and avian pathogenic *E. coli* [APEC] χ7122, its plasmid cured derivative χ7368, *E. coli* K-12 strain χ6092, *E. coli* K-12 strain MG1655 spontaneous nalidixic acid resistant mutant, and human commensal *E. coli* HS-4 spontaneous rifampicin mutant as recipients (Table 1). Strains were streaked onto MacConkey agar containing 4 mg/L cefotaxime (ESBL donors), 30 mg/L nalidixic acid (χ7122, χ7368, and MG1655), 100 mg/L rifampicin (HS-4), and 15 mg/L tetracycline (χ6092). Single colonies were then suspended in 3 mL of LB broth supplemented with appropriate antibiotics for either the donor or recipient and shaken overnight at 37 °C. The overnight cultures were converted to OD600 ~ 1.0 with LB broth (0.1% glucose) and pelleted by centrifugation at 10,000× *g* for 10 min. Donor and recipient pellets were resuspended with fresh LB, mixed in a 1:1 ratio, and incubated overnight at 37 °C. The following day, conjugation mixtures were serially diluted and plated on MacConkey agar with 4 mg/L cefotaxime (donor), 30 mg/L nalidixic acid (χ7122 and χ7368), 100 mg/L rifampicin (HS-4), or 15 mg/L tetracycline (χ6092), and 4 mg/L cefotaxime, 30 mg/L nalidixic acid, 100 mg/L rifampicin, or 15 mg/L tetracycline (transconjugants). Transconjugants were then subjected to confirmation via plasmid gel profiling and PCR replicon typing.

### 2.6. Statistics

Prism software version 6.0 (GraphPad, San Diego, CA, USA) was used to calculate significance for all statistical analyses. One-way ANOVA followed by Turkey’s test for multiple comparisons of means was used to compare differences between groups. *p* values < 0.05 were considered significant.

## 3. Results

### 3.1. Widespread Antibiotic Resistance Detected in Poultry Fecal Isolates

The breakdown of the total numbers of CRE, ESBL-producing *E. coli* and *Acinetobacter*, and CRA can be found in Table 2. Colistin resistant bacteria were detected, whereas ESBL-producing *E. coli* and *Acinetobacter* isolates were not detected until after enrichment. Colistin resistance was widely observed in bacteria from both CC and CF environments on CHROMagar COL-APSE™ plates (Figure 2). There were no significant differences (*p* > 0.05) in colistin-resistant bacteria abundances between the three laying periods within either environment. Of the total colonies that were resistant to colistin, 3.56% in CC and 0.69% in CF were identified as CRE (Table 2). A total of 52 CRE were isolated from CC and 48 from CF for future experiments. A total of 29 (60%) CRE in CC and 27 (47%) CRE in CF were shown to be resistant to colistin at concentrations up to 64 mg/L (Appendix A). Using CHROMagar Orientation™ plates post-enrichment, ESBL-production was relatively uncommon in both CC and CF isolates regardless of laying period. Of ESBL-producing isolates in the CC environment, 48% were identified as *E. coli* and 31% were identified as *Acinetobacter* spp. (Table 2). Of the ESBL-producing isolates from the CF environment, 3% were identified as *E. coli* and no *Acinetobacter* isolates were identified. Using CHROMagar SuperCARBA™ plates post-enrichment, carbapenem-resistant bacteria were solely identified in the CF environment. Of these isolates, 13% were identified as *Acinetobacter*. Importantly, the CRA isolates were shown to be resistant to both colistin and β-lactams, and were found in feces from CF hens in peak and late lay periods. All ESBL-producing *E. coli* and CRA were used for further experimentation. The minimum inhibitory concentration (MIC) of all CRE, ESBL-producing *E. coli*, and *Acinetobacter* to colistin, cefotaxime, and ceftazidime are listed in Appendix A.

Antibiotic resistance profiles for all CC and CF CRE, ESBL-producing *E. coli*, and *Acinetobacter* isolates are detailed in Figure 3 and Figure 4, respectively. Tetracycline resistance was widely distributed in 45 (69%) and 31 (56%) of AMR bacteria in CC and CF, respectively. A total of 20 CC and 13 CF *E. coli* isolates were MDR against a combination of three or more antibiotics. CC and CF *Acinetobacter* were shown to be MDR as they were resistant to all antibiotics tested. CC *Acinetobacter* were not able to grow on CHROMagar SuperCARBA™ media like their CF counterparts (Figure 3 and Figure 4). Using PCR to identify AMR genes corresponding to resistance mechanisms in these isolates, we found no isolates carrying any of the *mcr* genes tested. Additionally, all ESBL-producing CC and CF *E. coli* carried the *bla_CMY_* gene, which encodes the Class C β-lactamase AmpC (Figure 5). Furthermore, only CC AmpC-producing *E. coli* carried the *bla_TEM_* gene, which encodes the Class A ESBL TEM. The MDR *Acinetobacter* from CC also carried the *bla_CMY_* gene. However, CRA isolates were tested negative for the β-lactamase and carbapenemase-expressing genes tested in this study.

### 3.2. Multiple Plasmid Types Found in AMR E. coli Isolates

AMR *E. coli* isolates from CC and CF environments contained a variety of plasmids, ranging in size from 6 kb to 150 kb (Figure 6). However, we did not successfully extract plasmids from *Acinetobacter* isolates under any conditions.

Multiple incompatibility types, including I1, N, HI1, B/O, L/M, P, Y, FIA, FIB, and FIC, were found in *E. coli* isolates across all laying periods in both environments (Figure 3 and Figure 4). IncFIB plasmids were detected in 38 (58%) and 27 (49%) of CC and CF AMR *E. coli* respectively (Figure 3 and Figure 4). Furthermore, ESBL-producing *E. coli* from the CC environment all carried similar plasmids identified as IncK/B plasmids (Figure 3). However, ESBL-producing *E. coli* from CF did not yield any identified plasmids or replicon types (Figure 4). Replicon types IncA/C, FIIA, T, Frep, X, W, and HI2 were not detected in any isolates in this study.

### 3.3. Various ExPEC Phylogenetic Groups and Virulence Markers Genes Identified in AMR E. coli Isolates

Using PCR to predict virulence potential of AMR *E. coli* isolates, we found the ExPEC virulence factors *iutA* and *iss* to be widespread in CC (Figure 3) and CF (Figure 4) AMR *E. coli*. A total of 28 (48%) and 20 (34%) of CC AMR *E. coli* carried *iutA* and *iss*, respectively. Similarly, a total of 38 (69%) and 17 (31%) of CF AMR *E. coli* carried *iutA* and *iss*, respectively. Furthermore, *E. coli* phylogenetic typing yielded both B2 and D groups known for their virulence and relation to ExPEC (Figure 3 and Figure 4) [39]. Specifically, 8 (14%) AMR *E. coli* were identified as B2 and 10 (17%) as D in the CC environment. Similarly, 12 (24%) AMR *E. coli* were identified as B2 and 4 (8%) as D in the CF environment.

### 3.4. bla_CMY_ Can Be Exchanged between Virulent and Non-Virulent E. coli through an IncK/B Plasmid

Using in vitro conjugation assays to uncover a mechanism for β-lactam resistance dissemination, ESBL-producing *E. coli* (IA-EC-0010 and IA-EC-0018) isolates from CC were able to transfer the resistance of both cefotaxime and ceftazidime to both APEC strain χ7122 and its plasmid-cured derivative χ7368, but not to *E. coli* K-12 strain χ6092, MG1655, or human commensal HS-4. The AMR transfer was linked with the consistent transfer of a large 106 kb IncK/B plasmid (pIA-EC-0018-2) carrying the *bla_CMY_* gene (Figure 5). However, using χ7122 (pIA-EC-0018-2) transconjugants from the previous assay as donors resulted in successful transfer of this large plasmid to χ6092 (Figure 7). The virulence markers *iss* and *iutA* that were observed in the donor strains were not transferred to the recipients. CF *E. coli* were unable to transfer *bla_CMY_* under any conditions tested in this study.

## 4. Discussion

Commensal bacteria like *Enterobacteriaceae* and pathogens isolated from the chicken GIT are commonly resistant to multiple antibiotics [40]. Commonly, *E. coli* isolates sourced from chickens have shown resistance to tetracycline, chloramphenicol, ampicillin, and streptomycin due to the indiscriminate use of these antibiotics as growth promoters in poultry production [41,42]. Concurrent with these studies, our results show the extensive antibiotic resistance patterns associated with *E. coli* from both CC and CF farms in all maturity stages. Furthermore, the amount of CRE isolated from each maturity stage did not differ. Alarmingly, numerous CRE isolates from both CC and CF showed MDR to multiple antibiotics tested. Recent research has shown that the maturity of layer hens plays a significant role in the diversity of the ceca microbiota [26]. Although the detection of CRE is increasing in the poultry environment [15,43,44], there is little information on how maturity can impact the presence of these AMR bacteria. Factors such as disease and antibiotic intervention can drive the emergence of AMR bacteria in the poultry environment [45]. As the presence of MDR bacteria from chickens increases, the detection of specific resistance genes illustrates a relationship between resistant bacteria and the spread of resistance mechanisms. Future studies should investigate the role that the environment and possible intervention strategies that can limit the emergence of colistin resistant bacteria.

Our study highlights the importance of detection methods for colistin resistant bacteria using media like CHROMagar COL-APSE™. Notably, CHROMagar COL-APSE™ adheres to the European Committee on Antimicrobial Susceptibility Testing (EUCAST). The EUCAST breakpoint for colistin resistance is strictly set at 2 mg/L, whereas CLSI resistance cutoff in the United States is 4 mg/L. All isolates that were initially selected from CHROMagar COL-APSE™ were identified as resistant on MHA according to the EUCAST breakpoint of 2 mg/L [46]. The number of isolates initially selected from the CHROMagar COL-APSE™ media was greatly reduced after screening using MHA with 4 mg/L colistin. Using both CHROMagar COL-APSE™ media and further selection using CLSI guidelines can be useful for accurate detection of resistant isolates. Although use of CHROMagar COL-APSE™ and further confirmation with either CLSI or EUCAST guidelines can identify multiple resistant bacteria, these methods do not accurately detect the presence of *mcr* genes [47]. PCR investigation of all confirmed colistin resistant bacteria should be performed to confirm the presence of the emerging *mcr* genes.

The emergence of *mcr* genes in multiple settings have become an increasing global threat [14,48]. Notably, the principle *mcr-1* gene has been reported in over 40 countries to date [49]. None of the *mcr* genes tested in our study (*mcr-1*, *mcr-2*, *mcr-3*, *mcr-4*, and *mcr-5*) were detected in any of our CRE isolates. Although our isolates did not carry the *mcr* genes that are often spread through conjugative plasmids [50], our study highlights the ability of AMR *E. coli* from chickens to exhibit spontaneous resistance to colistin. There are several mechanisms that can contribute to the ability of bacteria to become spontaneously resistant to colistin. For instance, the most common resistance strategy involves the modification of the bacterial outer membrane through alteration in the lipopolysaccharide (LPS) and reduction in its negative charge as this negative charge is the target of colistin [51]. Furthermore, overexpression of efflux-pump systems and overproduction of capsule polysaccharides can enhance the resistance to colistin in *Enterobacteriaceae* [52,53]. The possibility of threatening AMR bacteria like ESBL-producing *E. coli* and MDR *Acinetobacter* to exhibit resistance to colistin spontaneously is concerning because of colistin’s use as a last resort antibiotic [54].

As evidenced in this study, the detection of ESBL-producing *E. coli* and MDR *Acinetobacter* can be detected in the poultry environment. In order to understand the genetic attributes that contributed to the resistance of these isolates, we investigated multiple β-lactamase and carbapenemase producing genes in both groups. Specifically, we attempted to detect the *bla_TEM_*, *bla_SHV_*, *bla_CTX_*_-*M*_, and *bla_CMY_* β-lactamase genes as these β-lactamases are the most commonly identified [55]. Furthermore, we attempted to detect carbapenemase genes *bla_OXA_*, *bla_KPC_*, and *bla_VIM_* that are commonly found in CRA [56,57,58]. As none of the β-lactamase and carbapenemase genes were detected in the CRA isolates, other resistance mechanisms are likely responsible for the resistance. Carbapenem resistance in *Acinetobacter* can be mediated by reduced drug permeability through porin loss or modification and the overexpression of efflux pump [59,60]. Interestingly, we successfully identified the AmpC Class C β-lactamase encoding *bla_CMY_* gene in only the early lay period of both CC and CF environments. *bla_CMY_* has been increasingly reported in chickens as a source for the spread of AmpC to both human and avian pathogenic bacteria [61,62]. Although *bla_CMY_* is not classified as an extended-spectrum β-lactamase, the AmpC β-lactamase is resistant to otherwise useful β-lactamase inhibitors like clavulanic acid that is often paired with antibiotics, like ceftazidime, to limit the effects of β-lactamases [63,64]. Strikingly, in our study the ESBL gene *bla_TEM_* was detected in all cephalosporin resistant CC *E. coli*, illustrating the ability of AMR *E. coli* to harbor multiple β-lactamase producing genes.

To evaluate whether CRE and ESBL-producing *E. coli* detected in this study can be pathogenic, we tested our isolates using *E. coli* phylogenetic typing, a common technique utilized to sort *E. coli* isolates into groups that differ in ecological niches, life-history characteristics, and propensity to cause disease [65]. Namely, four main *E. coli* phylotypes (A, B1, B2, and D) are extensively used for classification [66,67,68]. We identified A, B2, and D groups in all maturity stages in both environments. Historically, phylogenetic group A is more commonly associated with commensal *E. coli*, whereas phylogenetic groups B2 and D are associated with virulent ExPEC infections [69,70]. For instance, phylogenetic groups B2 and D have been associated with *E. coli* isolates that cause urinary tract infections (UTI) and show increased presence of virulence [71] factors associated with UTI when compared to the B1 and A phylogenetic groups.

Additionally, because of their plasmidic location, we attempted to detect four ExPEC-associated virulence factors (*cvaC*, *iroN*, *iss*, and *iutA*) among our isolates [33]. *cvaC* and *iroN* were not identified in any of our isolates. However, there was wide distribution of both *iss* and *iutA* in isolated from both environments. Interestingly, the late lay period of the CC isolates exhibited markedly less *iss* and *iutA* than the early and peak lay periods. Encoding the receptor for the siderophore aerobactin, *iutA* is an important factor in urinary pathogenic *E. coli* (UPEC) infections that allow the bacteria to competitively acquire *iroN* that would otherwise be acquired by the host [72]. Furthermore, the *iss* virulence factor encodes a specific outer membrane protein that increases serum survival for ExPEC during extraintestinal infection [73]. In addition to their role in the pathogenesis, these virulence factors could confer a competitive advantage to isolates that express them compared to their counterparts in the GIT.

The ability for *E. coli* to spread AMR genes through plasmids via HGT plays a significant role on the distribution of resistance in different bacteria. Several factors play a role in the ability of plasmids to be transferred between bacteria. For instance, incompatibility restriction, host genetics, and strain-specific factors can influence the transfer of plasmids [38,74,75]. In this study, we investigated a total of 18 plasmid incompatibility groups in all *E. coli* isolates. We were able to identify several isolates that carry IncN, P, HI1, and I1 incompatibility groups that have been shown to be associated with a broad-host range [76,77,78]. Interestingly, in our study, all ESBL-producing *E. coli* in the CC environment carried IncK/B type plasmids. Due to their multiple plasmids visualized in plasmid extraction, and presence of the common IncK/B plasmids, the CC ESBL-producing *E. coli* were selected for conjugation assays. CF *E. coli* that carried the *bla_CMY_* gene did not yield any of the plasmid replicon types investigated in this study, and conjugation was not observed under any conditions. Furthermore, CRA isolates did not yield any plasmid replicon types investigated in this study, and no plasmids were visualized using the method described or using IBI-Scientific (Dubuque, IA, USA) and Qiagen (Hilden, Germany) commercially available plasmid extraction kits. This suggests that the AMR genes of these isolates could be chromosomal rather than plasmidic.

In this study, we demonstrated the ability of the ESBL-producing *E. coli* from CC to transfer AmpC β-lactamase via a large 106 kb IncK/B plasmid (pAmpC) to a variety of hosts, including the APEC strain χ7122, its avirulent plasmid-cured derivative χ7368, and *E. coli* K-12 χ6092, indicating the ability of the plasmid to transfer to both virulent and non-virulent bacteria. Interestingly, numerous studies have highlighted the presence of pAmpC associated with both IncK plasmids and the poultry environment [79,80,81,82]. Although transfer of AMR genes and plasmidic virulence factors on the same plasmid has been shown [83,84], the ESBL-producing donors in our study did not transfer the virulence factors *iss* and *iutA* to the avirulent strains. This suggests that the plasmidic virulence factors are located on different plasmids than the pAmpC in the donor strains. Nonetheless, the presence of pAmpC in populations of MDR bacteria with the propensity to cause disease is concerning. Finally, increased detection of pAmpC can cause an eventual decline in the efficacy of cephalosporin antibiotics in both humans and poultry.

## 5. Conclusions

Overall, this study identified wide-spread colistin-resistant *E. coli* and MDR *E. coli* in all three lay periods in CC and CF hens. AmpC β-lactamase mediated *bla_CMY_* was identified in *E. coli* and *Acinetobacter* of CC and *E. coli* of CF. Furthermore, MDR *Acinetobacter* were detected in peak and late lay periods of both CC and CF, with CF isolates being also resistant to carbapenems. Finally, the AmpC β-lactamase mediated *bla_CMY_* was shown to be plasmid-borne in the CC environment only. Future experimental studies will seek environmental and host factors that trigger dissemination of novel and threatening antibiotic resistant patterns to enable comparisons between CC and CF hens, as well as further sequencing of resistant isolates and transferable plasmids to understand the prevalent resistance genes in laying hens of different maturity stages.

## Figures and Tables

**Figure 1 microorganisms-09-00141-f001:**
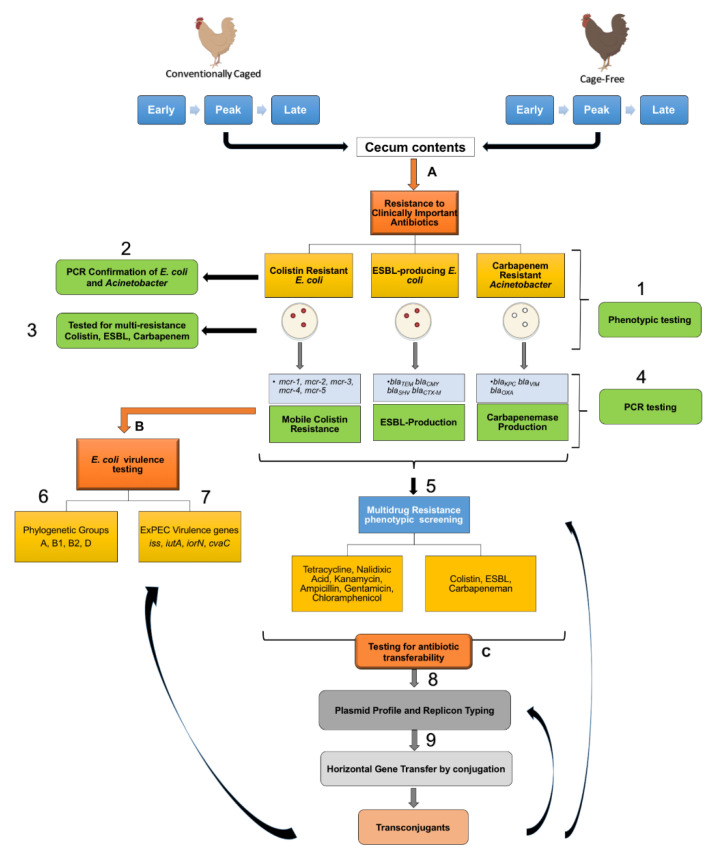
Graphic description of sample analysis. Ceca contents from hens at different maturity stages (early, peak, and late) from conventional cage and cage-free poultry farms were analyzed for (**A**) resistance to clinically important antibiotics, (**B**) *E. coli* virulence genes and phylotype, and (**C**) plasmid profiles and transferability.

**Figure 2 microorganisms-09-00141-f002:**
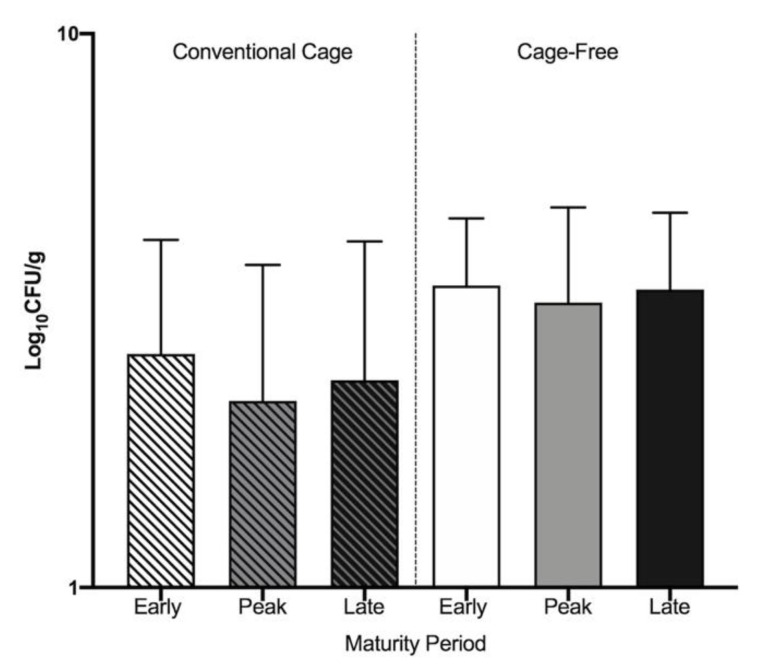
Prevalence of colistin-resistant bacteria in CC and CF. Total colistin-resistant bacteria were counted and Log_10_ CFU/g were calculated for each lay period in both environments.

**Figure 3 microorganisms-09-00141-f003:**
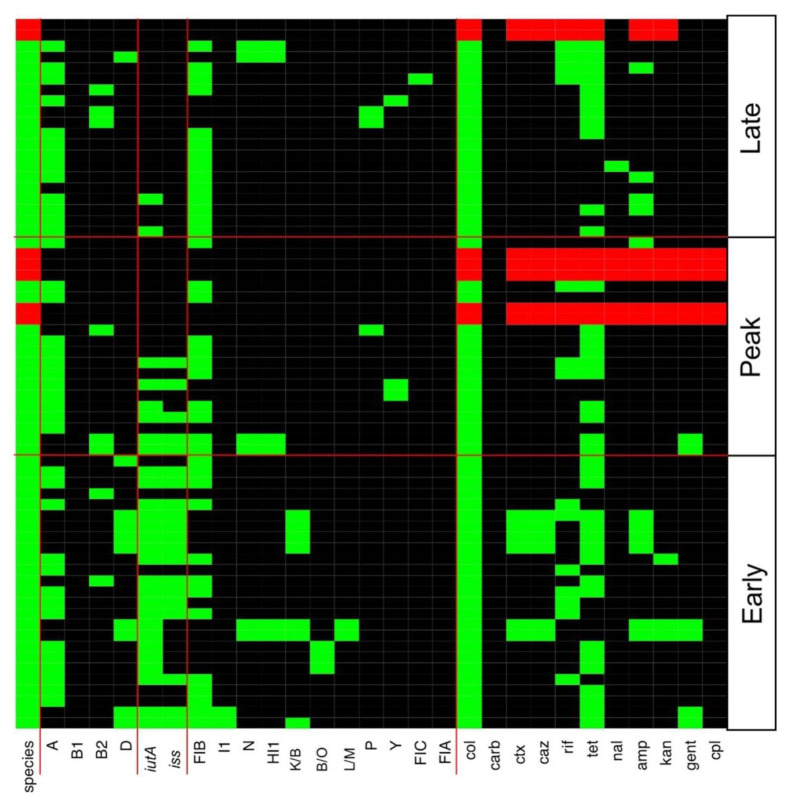
Heatmap of genes and phenotypes detected in conventional cage antibiotic resistant isolates. All positive *E. coli* results are shown in the heatmap as green. All positive *Acinetobacter* spp. are shown as red. Negative results are shown in black in the heatmap. Col, colistin; carb, carbapenem; ctx, cefotaxime; caz, ceftazidime; rif, rifampicin; tet, tetracycline; nal, nalidixic acid; amp, ampicillin; kan, kanamycin; gent, gentamicin; cpl, chloramphenicol.

**Figure 4 microorganisms-09-00141-f004:**
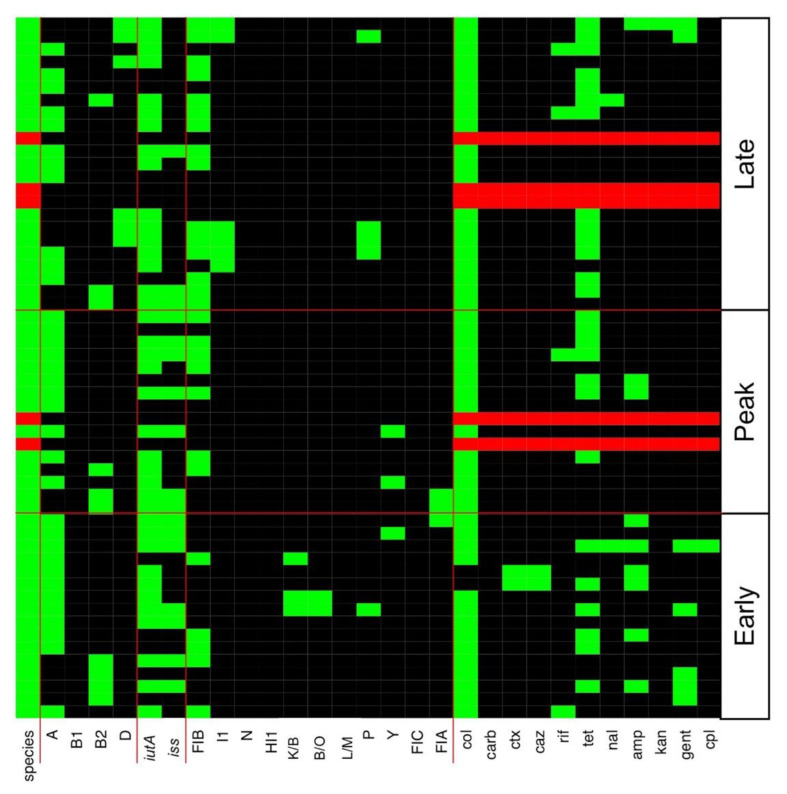
Heatmap of genes and phenotypes detected in cage-free antibiotic resistant isolates. All positive *E. coli* results are shown in the heatmap as green. All positive *Acinetobacter* spp. are shown as red. Negative results are shown in black in the heatmap. Col, colistin; carb, carbapenem; ctx, cefotaxime; caz, ceftazidime; rif, rifampicin; tet, tetracycline; nal, nalidixic acid; amp, ampicillin; kan, kanamycin; gent, gentamicin; cpl, chloramphenicol.

**Figure 5 microorganisms-09-00141-f005:**
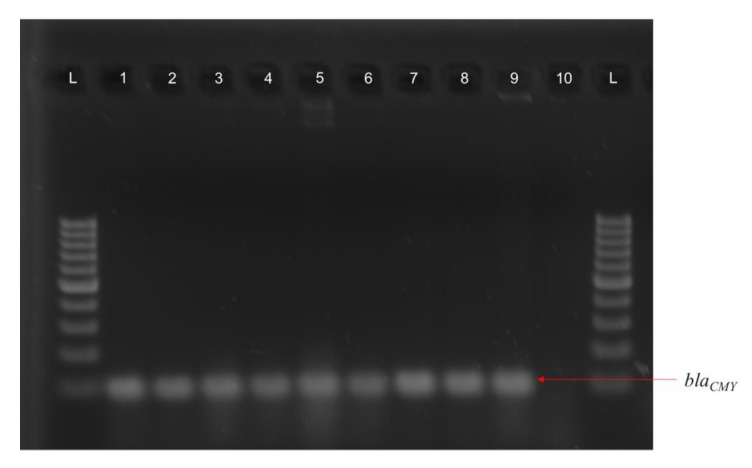
PCR for *bla_CMY_* in conventional cage and cage-free *E. coli* isolates. (**L**) 100 bp ladder, (**1**) IA-EC-0010; (**2**) IA-EC-0011; (**3**) IA-EC-0018; (**4**) IA-EC-0019; (**5**) IA-EC-0020; (**6**) IA-EC-0021; (**7**) IA-EC-0075; (**8**) IA-EC-0076; (**9**) χ7122 (pIA-EC-0018) transconjugant; (**10**) χ7122 negative control.

**Figure 6 microorganisms-09-00141-f006:**
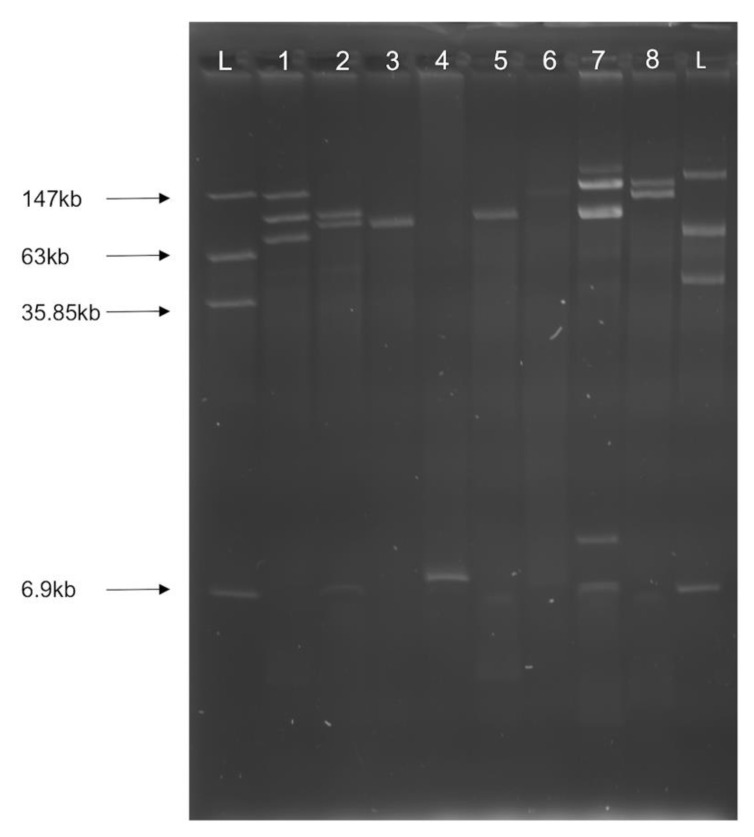
Plasmid profiles of representative AMR *E. coli* isolates. (**L**) Ladder *E. coli* 39R681; (**1**) IA-EC-0002; (**2**) IA-EC-0024; (**3**) IA-EC-0030; (**4**) IA-EC-0035; (**5**) IA-EC-0052; (**6**), IA-EC-0060; (**7**) IA-EC-0010; (**8**) IA-EC-0073. Plasmid size was approximated using GelAnalyzer software.

**Figure 7 microorganisms-09-00141-f007:**
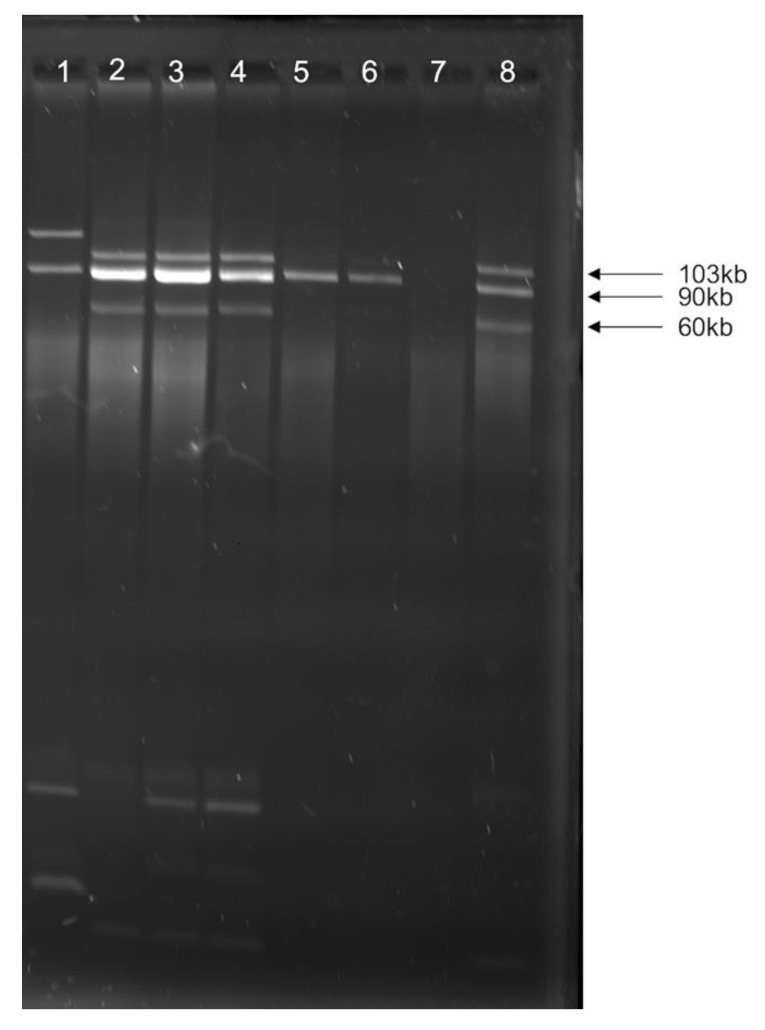
Plasmid profiles of donor, recipient, and transconjugant strains. (**1**) Donor IA-EC-0018; (**2–4**) transconjugant χ7122 (pIA-EC-0018-2); (**5–6**) trans-conjugant χ6092 (pIA-EC-0018-2); (**7**) recipient χ6092; (**8**) recipient χ7122.

**Table 1 microorganisms-09-00141-t001:** Relevant characteristics of donor and recipient strains.

*E. coli* Strain	Notable Characteristics	Selected Plasmidic Virulence Factors	Source
**Recipients**			
χ7122	APEC O78:K80:H9, nal^R^, str^R^, lac^+^	*cvaC*, *iss*, *iutA*, *iroN*	[35]
χ7368	Plasmids-cured χ7122: ΔpChi7122-1 ΔpChi7122-2 ΔpChi7122-3, nal^R^, lac^+^	−	[33]
χ6092	*E. coli* K-12, tet^R^, lac^−^	−	[36]
MG1655	*E. coli* K-12, nal^R^, lac^+^	−	[37]
HS-4	Human commensal *E. coli*, rif^R^, lac^−^	−	[38]
**Donor**			
IA-EC-0010 (CC)	Lac^+^, ctx^R^, caz^R^, col^R^	*iutA*	This Study
IA-EC-0018 (CC)	Lac^+^, ctx^R^, caz, col^R^	*iss*, *iutA*	This Study
IA-EC-0075 (CF)	Lac^+^, ctx^R^, caz^R^	*iutA*	This Study
IA-EC-0076 (CF)	Lac^+^, ctx^R^	*iutA*	This Study
**Transconjugants**			
χ7122 (pIA-EC-0010-2)	nal^R^, str^R^, ctx^R^, caz^R^ lac^+^, pIA-EC-0010-2 (Donor IA-EC-0010)	*cvaC*, *iss*, *iutA*, *iroN*	This Study
χ7368 (pIA-EC-0010-2)	nal^R^, ctx^R^, caz^R^, lac^+^, pIA-EC-0010-2 (Donor IA-EC-0010)	−	This Study
χ7122 (pIA-EC-0018-2)	nal^R^, str^R^, ctx^R^, caz^R^ lac^+^, pIA-EC-0018-2 (Donor IA-EC-0018)	*cvaC*, *iss*, *iutA*, *iroN*	This Study
χ7368 (pIA-EC-0018-2)	nal^R^, ctx^R^, caz^R^, lac^+^, pIA-EC-0018-2 (Donor IA-EC-0018)	−	This Study
χ6092 (pIA-EC-0018-2)	tet^R^, ctx^R^, caz^R^,lac^−^, pIA-EC-0018-2 (Donor χ7122 (pIA-EC-0018-2)	−	This Study

Notes: nal, nalidixic acid; str, streptomycin; tet, tetracycline; ctx, cefotaxime; caz, ceftazidime; col, colistin; rif, rifampicin; R, resistance; lac, lactose fermentation; + or − positive or negative; CC, conventional cage; CF, cage-free.

**Table 2 microorganisms-09-00141-t002:** Antimicrobial resistant bacteria detected in CC and CF Environments.

Environment	Lay Period	Positively Identified Antibiotic Resistant Isolates						
		Pre-Enrichment		Post-Enrichment				
		**Total** **Colistin^R^ (CFU/g)**	**CRE** **CFU/g (% Total)**	**Total** β **-Lactam^R^ (CFU/g)**	β **-Lactam^R^** ***E. coli* CFU/g (% Total)**	β **-Lactam^R^** ***Acinetobacter* CFU/g(% Total)**	**Total Carba^R^ (CFU/g)**	**CRA CFU/g (% Total)**
**CC**	Early	3.7 × 10^3^	2 × 10^2^ (5.57)	41.7	41.7 (100)	0	0	0 (0)
	Peak	1.9 × 10^3^	1 × 10^2^ (6.21)	55.6	0 (0)	41.5 (76.4)	0	0 (0)
	Late	7 × 10^3^	1 × 10^2^ (1.67)	5.6 10^2^	0 (0)	13.3 (36.2)	0	0 (0)
**CF**	Early	2 × 10^4^	1 × 10^2^ (0.52)	1.7 10^2^	17.9 (10.5)	0 (0)	50.5	0 (0)
	Peak	2 × 10^4^	1 × 10^2^ (0.57)	2 10^2^	0 (0)	0 (0)	1.3 × 10^2^	22.5 (16.7)
	Late	1 × 10^4^	1.5 × 10^2^ (1.19)	1.5 10^2^	0 (0)	0 (0)	4 × 10^2^	25.9 (6.3)

Notes: R, resistance; CC, conventional cage; CF, cage-free; CFU, colony forming unit; CRE, colistin resistant *E. coli*; Carba, carbapenem; CRA, carbapenem resistant *Acinetobacter*.

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
