# Peer review of "Bacteria Broadly-Resistant to Last Resort Antibiotics Detected in Commercial Chicken Farms"

_microorganisms, 2021, doi:10.3390/microorganisms9010141_

Round 1
Reviewer 1 Report
The study by Jochum et al. investigates the extend of MDR bacteria in poultry. Their study is carried out carefully and looks at many aspects of antibiotic resistance such as mechanism of resistance, their dissemination and connection to virulence factors, in terms of co-transfer to other bacterial populations.
It is without any doubt that presence of antibiotic resistance genes in poultry is major concern. So this is a relevant study.
Minor comments:
As a person not familiar with poultry I could not understand the relationship and the significance of maturity layers in MDR bacteria. These layers have to be explained.
Also, the authors report that CRA in Acinetobacter was not due to beta-lactamases of carbapenemase-expressing genes, lines 213-214. What is the CRA due to?
In addition, no discussion is extended for the observation that no plasmid from Acinetobacter species could be extracted. Does this mean that antibiotic resistance is chromosome encoded? Some discussion on this point is worth making.
Ceca or cecum contents? The authors have to select the one that is correct. In the abstract it is written “cecum contents”, but in the rest of the manuscript is written “ceca contents”, which is the correct form.
Line 294, it should read “…highlights” not “highlighted”
Line 359, the word “do” is missing.
Reviewer 2 Report
The manuscript describes the microbiological characterization of drug-resistant strains of bacteria isolated from poultry farms in Iowa, US, using 2 groups of animals: conventional-cage and cage-free. The distinction is important as farm use of antibiotics is linked to increased drug resistance in humans. Surprisingly, the data shows no statistical difference between the 2 groups. The only trend seen in the heat maps of drug resistance is an increase in the number of drug-resistant strains with the age of hens. The finding is understandable considering an increased time to transfer the plasmid within an animal group.
Presentation, discussion, and graphs are very good except for the heat maps. They seem to be prepared from a low-resolution figure and the quality has to be improved for publication.
